# Potent Activity of a High Concentration of Chemical Ozone against Antibiotic-Resistant Bacteria

**DOI:** 10.3390/molecules27133998

**Published:** 2022-06-22

**Authors:** Karyne Rangel, Fellipe O. Cabral, Guilherme C. Lechuga, João P. R. S. Carvalho, Maria H. S. Villas-Bôas, Victor Midlej, Salvatore G. De-Simone

**Affiliations:** 1Center for Technological Development in Health (CDTS), National Institute of Science and Technology for Innovation in Neglected Population Diseases (INCT-IDPN), FIOCRUZ, Rio de Janeiro 21040-900, Brazil; gclechuga@gmail.com (G.C.L.); joaopedrorsc@gmail.com (J.P.R.S.C.); 2Laboratory of Epidemiology and Molecular Systematics (LESM), Oswaldo Cruz Institute, FIOCRUZ, Rio de Janeiro 21040-900, Brazil; 3Microbiology Department, National Institute for Quality Control in Health (INCQS), FIOCRUZ, Rio de Janeiro 21040-900, Brazil; fellipe.cabral@incqs.fiocruz.br (F.O.C.); maria.villas@incqs.fiocruz.br (M.H.S.V.-B.); 4Post-Graduation Program in Science and Biotechnology, Department of Molecular and Cellular Biology, Biology Institute, Federal Fluminense University, Niterói 22040-036, Brazil; 5Laboratory of Cellular and Ultrastructure, Oswaldo Cruz Institute, FIOCRUZ, Rio de Janeiro 21040-900, Brazil; victor.midlej@ioc.fiocruz.br

**Keywords:** ozone, pathogenic bacteria, antimicrobial resistance, SEM, ESKAPE pathogens, antimicrobial activity

## Abstract

Background: Health care-associated infections (HAIs) are a significant public health problem worldwide, favoring multidrug-resistant (MDR) microorganisms. The SARS-CoV-2 infection was negatively associated with the increase in antimicrobial resistance, and the ESKAPE group had the most significant impact on HAIs. The study evaluated the bactericidal effect of a high concentration of O_3_ gas on some reference and ESKAPE bacteria. Material and Methods: Four standard strains and four clinical or environmental MDR strains were exposed to elevated ozone doses at different concentrations and times. Bacterial inactivation (growth and cultivability) was investigated using colony counts and resazurin as metabolic indicators. Scanning electron microscopy (SEM) was performed. Results: The culture exposure to a high level of O_3_ inhibited the growth of all bacterial strains tested with a statistically significant reduction in colony count compared to the control group. The cell viability of *S. aureus* (MRSA) (99.6%) and *P. aeruginosa* (XDR) (29.2%) was reduced considerably, and SEM showed damage to bacteria after O_3_ treatment Conclusion: The impact of HAIs can be easily dampened by the widespread use of ozone in ICUs. This product usually degrades into molecular oxygen and has a low toxicity compared to other sanitization products. However, high doses of ozone were able to interfere with the growth of all strains studied, evidencing that ozone-based decontamination approaches may represent the future of hospital cleaning methods.

## 1. Introduction

Health care-associated infections (HAI) are a significant public health problem worldwide, especially in developing countries where the frequency can be at least three times higher than that of resource-rich countries [1]. It is estimated that approximately four million people acquire HAIs in the European Union (EU) and that some 37,000 persons die due to resistant infections acquired in hospital environments. Most of these deaths (67.6%) are caused by multidrug-resistant (MDR) bacteria to antimicrobials [2]. The 2016 European Annual Report recorded the incidence density (DI) for ventilator-associated pneumonia (VAP) of 3.9/1000 days, central catheter-associated bloodstream infections (CCAB) of 1.7/1000 days, and infections of the urinary tract related to a catheter (ITURC) of 2.1/1000 days [2], while in Brazil, the DI of device-related HAIs in the year 2016 indicated VAP of 13.6/1000 days, primary clinical bloodstream infection associated with central vascular catheter (CBIACC) of 4.6/1000 days, and catheter-related urinary tract infections (CRUTI) of 5.1/1000 days [3]. The increasing burden of HAIs stemming from poor infection monitoring and control practices are among the drivers of antimicrobial resistance. Evidence indicates a strong relationship between antimicrobial resistance and HAIs [4], with MDR pathogens being a common cause [5,6]. Although they are frequent adverse events with high morbidity and mortality rates and costs, HAIs are recognized as preventable in up to 70% of cases [7].

Antimicrobial resistance (AMR) results from bacteria’s natural evolution and adaptation processes. However, selection pressure has accelerated it, originating from the inadequate or excessive use of antimicrobials, favoring MDR microorganisms’ appearance and rapid spread [8,9,10]. The problem is highlighted in the ICU, where patients have higher risk factors for nosocomial infections. In addition, the cost of antimicrobial resistance in these infections is very high, as diseases caused by these pathogens have worse clinical outcomes, prolonged hospital stays, and increased mortality rates [11]. More than 700,000 deaths are associated with AMR [12,13], and by 2050, the number of lives lost annually could reach 10 million [14]. The COVID-19 pandemic was declared by the World Health Organization (WHO) on 12 March 2020 [15,16,17]; at that time, the disease had been spreading rapidly since the first detection of the SARS-CoV-2 coronavirus in Wuhan, China, in December 2019 [18]. The SARS-CoV-2 infection had a negative association with the increase in antimicrobial resistance for reasons related mainly to the rise in the practical use of antimicrobials, the overcrowding of health systems, a lack of management measures, and a decrease in the pace of activity of laboratories in surveillance cultures and diagnostic tests to detect antimicrobial-resistant organisms.

On the other hand, the lower impact on the development of antimicrobial resistance may be associated with increased infection control measures adopted to prevent the contamination of healthcare professionals with SARS-CoV-2, including hand hygiene and the use of individual protective equipment and devices to decontaminate the air and surfaces [19]. According to some studies, up to 5% of patients infected with SARS-CoV-2 had to be admitted to the ICU [20,21]. In addition, it has been documented that up to 50% of these patients may have had secondary bacterial infections or superinfections, mainly bacteremia and urinary tract infections [22,23]. Undoubtedly, the dramatic increase in COVID-19 deaths includes HAI coinfection cases. Furthermore, the hospitalization length increases the risk of being affected by HAIs, which may even exacerbate a severe morbidity condition, further leading to the patient’s death, particularly if with comorbidity [24]. A study in Lombardy, the Italian region with the most COVID-19 deaths, revealed that most HAIs occurred in ICUs [25]. Other authors reported the high prevalence of HAIs in ICUs in Italy and associated such infections with the use of a urinary catheter, surgical drainage, intravascular catheters, and mechanical ventilation [26,27]. These infections are more prevalent in terminally ill patients and are primarily due to the spread of MDR pathogens.

Among the MDR pathogens, those from the ESKAPE group have the most significant impact on HAIs. Also called “super bacteria”, they group six pathogens that can escape the biocidal activity of antimicrobials: *Enterococcus faecium*, *Staphylococcus aureus, Klebsiella pneumoniae*, *Acinetobacter baumannii*, *Pseudomonas aeruginosa*, and *Enterobacter* spp. [28,29]. The inefficiency of antimicrobials against these pathogens is due to several resistance mechanisms, such as drug inactivation, modification of drug binding sites/targets, changes in cell permeability, and mutation [30]. As a result, these pathogens can survive in the hospital environment for extended periods and be transported from one individual to another, thus spreading in the community and hospital [31]. 

A priority list of antimicrobial-resistant bacteria was described in 2017 by WHO to support renewed efforts in researching and developing new antimicrobials, diagnostics, vaccines, and other tools [32]. Most ESKAPE pathogens appear on this list of the most problematic microbial species, which appeals to focus research efforts on this topic [33]. The European Centre for Disease Prevention and Control (ECDC) and the US Centers for Disease Control and Prevention (CDC) provided the following standardized definitions for MDR, extensively drug-resistant (XDR), and pan-drug resistant (PDR) bacteria. MDR bacteria are defined as those with acquired resistance to at least one agent in three or more categories of antimicrobials. XDRs are not susceptible to at least one agent in all classes of antimicrobials except two or fewer types (i.e., they remain sensitive to only one or two categories). Bacteria resistance to all agents in all antimicrobial types is called PDR [34]. The environment plays a central role in transmitting hospital-acquired pathogens and the pathogenesis of HAIs.

Many bacteria, especially MDR, can survive in the hospital environment for several months, particularly in areas close to patients. Among the factors that favor the contamination of the health services environment, we can mention the hands of health professionals in contact with surfaces; maintenance of damp, wet, and dusty surfaces; precarious conditions of coatings; and maintenance of organic matter [35]. The presence of dirt, mainly organic matter of human origin, can serve as a substrate for the proliferation of microorganisms or favor the presence of vectors, which can passively carry these agents. This indicates the importance of rapid cleaning and disinfection of any area with organic matter, regardless of the hospital area [33]. The effective disinfection of surfaces and the environment is considered one of the primary measures to control the spread of HAIs. Unfortunately, many studies have concluded that current cleaning methods are microbiologically ineffective. This failure concerns daily cleaning and final cleaning after the patient is discharged. Improvements in environmental cleanliness are associated with a decrease in hospital-acquired pathogens and HAIs [36]. Last year, a new global emergency introduced the requirement for further disinfection and sanitization procedures to optimize the quality of care and work safety in professional environments [37,38].

According to all this and considering the increasing prevalence of MDR microorganisms in hospitals, which has become a severe threat to public health, the need for safe and validated technologies capable of ensuring the disinfection of air environments, room surfaces, and sanitary materials has become evident against the current pandemic or future events. In this sense, the study of alternative methods and/or agents for disinfection and sanitization should receive special attention, and ozone (O_3_) can be a valid option with different objectives [39]. Ozone is a blue-colored gas with a characteristic odor, presented in the triatomic form of oxygen (O_3_), and is partially soluble in water and highly unstable, decomposing quickly into oxygen. Therefore, it cannot be produced in large quantities without being continuously [40]. With an oxidative potential superior to most commercial disinfectants and a faster reaction faster than O_2_, it has been studied for decades in medicine and biological sciences, becoming a versatile therapeutic agent which helps treat several diseases [41]. The exposure to ozone, also known as the time concentration value (mg L-1 min), is the most important operational parameter in O_3_ disinfection, representing the time-integrated ozone concentration in its most general form. Medium temperature also has a strong influence [42,43]. The humidity is also a key parameter when ozone is applied in the gas phase, requiring high relative humidity conditions to obtain a significant inactivation of target microorganisms [44,45]. The chemical composition of the surface to be treated and its shape and texture could also be important factors. 

Gaseous disinfectants are proven to be effective because, in addition to the antimicrobial effect, gas can reach surfaces difficult to get by conventional cleaning [46,47]. For example, ozone gas in the concentration of 25 ppm caused a significant reduction in the number of viable bacteria and the total biomass of *K. pneumoniae* biofilm [48]. Furthermore, ozone seems to be very effective against planktonic bacteria, which are susceptible to ozone action and are often significantly reduced or completely eradicated from the surfaces with smaller concentrations [49,50,51,52]. Currently, with the COVID-19 pandemic, ozone has been investigated as a possible preventive measure for the spread of infection [53], in hospital hygiene for disinfecting rooms [54], in viability on different surfaces [55], and as a therapeutic option in the treatment of patients [24,56]. 

Ozone acts first on the cell membrane as a disinfectant, reacting with glycoproteins, glycolipids, and nucleic acids. Then, microorganisms are inactivated by cell disruption due to the action of molecular ozone or free radicals during the decomposition of the gas [57]. Studies show that ozone influences the global polarity of the bacterial surface [58], involving mechanisms of lipid peroxidation [59] and the degradation of transmembrane proteins that control the flow of ions. Thus, cells rupture with a subsequent leakage of ions between the media, resulting in the microorganism’s death [60]. 

Despite having been used in the hospital environment for some time, little is known about the potential of this agent, especially in the Brazilian context, where studies on the subject are scarce. Therefore, from this perspective and due to the aspects reported, there was an interest in evaluating the bactericidal action of high concentration ozone gas on some reference bacteria used in the process of assessing the bactericidal activity of disinfectants and some bacteria from the ESKAPE group that have a high antimicrobial resistance profile.

## 2. Materials and Methods

### 2.1. Bacterial Strains

Standard strains (*Staphylococcus aureus* (ATCC 6538), *Salmonella enterica* subsp. enterica serovar *choleraesuis* (ATCC 10708), *Escherichia coli* (ATCC 25922), and *Pseudomonas aeruginosa* (ATCC 15442)) were obtained from the American Type Culture Collection (ATCC) (Plast Labor Ind. Com. EH Lab. Ltd.a, Rio de Janeiro, Brazil). Representative MDR strains of the ESKAPE group were also used, with four clinical strains isolated from HAIs, which were: methicillin-resistant *S. aureus* (MRSA), carbapenemase-producing *K. pneumoniae* (KPC+), *A. baumannii* PDR carrying the *bla*_OXA-23_ gene and representing one of the genotypes disseminated in Brazil (ST15/CC15), and an environmental strain of *P. aeruginosa* (XDR) from hospital effluent. These strains were kindly provided by Dr. Maria H. S. Villas-Bôas (National Institute for Quality Control in Health of the Oswaldo Cruz Foundation—INCQS/FIOCRUZ) and Dra. Catia Chaia de Miranda (Interdisciplinary Medical Research Laboratory, LIPMED, FIOCRUZ). These bacterial strains were initially cultivated according to the instructions of the ATCC, aliquoted, and stored in cryotubes containing tryptic soy broth (TSB, Difco) with 20% glycerol (*v*/*v*) and kept at −20 °C for later use. 

### 2.2. Ozone Generating and Monitoring

The ozone generating equipment (SANITECH O3-80-Sanitization, Astech Serv. and Fabrication Ltd.a., Petrópolis, Brazil) is adjustable from 10 to 80 ppm, and the capacity to treat the room air up to 1000 m^3^ (not habitable) was used. The environmental concentration of O_3_ emitted was monitored and measured using two portable electrochemical ozone detection modules (model ZE14-O3) (Zhengzhou Winsen Electronics Technology Co., Ltd., Honã, Zhengzhou, China). In addition, this equipment was coupled to a module containing a digital temperature and relative humidity sensor (model AM2302) (Guangzhou ASAIR Electronic Co., Ltd., Guangzhou, China). The ZE14-O3/AM2302 modules constantly monitored these three parameters during the experiment, with 2–3 s detection and simultaneous recording on a computer. The measurement was made with the ZE14-O3/AM2302 (Sensors 1 and 2) inserted directly inside each container, always on the first shelf.

### 2.3. Inoculation of the Test Surface

The strains were removed from the freezer stock culture for bacterial reactivation, sown in TSB, and incubated at 37 °C for 24 h. After the microorganisms were suspended in sterile 0.85% saline, the concentration of 10^8^ colony-forming units (CFU) mL^−1^ was determined with a densitometer (Densichek Plus, BioMérieux, Rio de Janeiro, Brazil). The successive dilutions (10^5^ and 10^4^ CFU mL^−1^) were made in the brain–heart infusion broth (BHI). One hundred microliter aliquots of each bacterial suspension (*S. aureus, S. enterica*, *E. coli*, *P. aeruginosa, S. aureus* (MRSA), *K. pneumoniae* (KPC+), *A. baumannii* (PDR), and *P. aeruginosa* (XDR)) in different concentrations (10^5^ and 10^4^ CFU mL^−1^) were plated in triplicate by spread plate on Triptona Soy Agar (TSA; DIFCO Laboratories Inc., Detroit, MI, USA) and incubated at 37 °C for 24 h. 

### 2.4. Ozone Treatment

The ozone generated was infused into two hermetically sealed containers, with a volume of approximately 1 m^3^ each (Appendix A). The plates inoculated with the different microorganisms were placed on each container’s shelves. After closing the lid of each container, we started the exposure to ozone using only one SANITECH O3-80-Sanitization ozone generator, producing ozone at a concentration of 80 ppm (maximum) (Appendix A). ATCC strains were exposed to ozone for 1, 10, 20, 30, and 40 min. According to the results obtained with the reference strains, we verified that the initial concentration of the inoculum (10^4^ or 10^5^ CFU/mL) had no significant interference in the colony count results, but rather the time of exposure to ozone presented the best impact at 40 min. As a result, the other strains (ESKAPE) were exposed to ozone at 10^5^ CFU/mL/40 min. The ozone-generating equipment takes time to reach its maximum concentration (ppm). For this reason, for each exposure time determined, a 2 min addition to the readings was considered after this initial time. After the exposure time, the container was opened, and the plates were closed and incubated at 37 °C for 24 h. As a positive control for the assay, we used plates with TSA containing the same bacterial suspensions but without exposure to ozone. These plates remained at room temperature and were incubated at 37 °C for 24 h, with the plates exposed to ozone. A plate containing only TSA was used as a negative control. The test was performed in triplicate. Colony counting was performed only on plates with many colonies from 0 to 300. 

### 2.5. Cell Viability

The cell viability was measured on a selected bacterial suspension of 10^5^ CFU mL^−1^ after 40 min exposure to O_3_ based on previous results (cell count—CFU mL^−1^). The entire previous experiment was performed again (at the defined concentration and time), and after 24 h of incubation, three distinct colonies from each plate were inoculated separately in a test tube containing TSB broth (Difco). As a positive control of the assay, we performed the same procedure with the plates that were not exposed to O_3_, where three distinct colonies of each dish were inoculated separately in a test tube containing TSB broth (Difco). Afterward, 100 μL of the bacterial suspension of each colony was transferred, in triplicate, to the wells of the 96-well microplate, which was incubated at 37 °C for 24 h. Each strain was tested in duplicate and bacterial growth was detected by adding 0.02% resazurin (7-hydroxy-phenoxazin-3-one 10-oxide; Sigma-Merck, St. Louis, MO, USA) with 1 h incubation [61]. Resazurin is a non-toxic, non-fluorescent blue reagent that, after enzymatic reduction, becomes highly fluorescent. This conversion occurs only in viable cells; as such, the amount of resorufin produced is proportional to the number of viable cells in the sample [62,63,64]. As a negative control, we used TSB broth, and the measurement at 590 nm was conducted on an ELISA plate reader (Flex Station 3; Molecular Devices, San José, CA, USA). 

The collected data were analyzed using the program R (version 3.6.0) (Vienna, Austria) and R Studio, where the paired t-test was applied to compare the statistical significance between the two samples (with and without treatment with O_3_) with ≤0.01. Each experiment was repeated three times for each microorganism treated with O_3_.

### 2.6. Scanning Electron Microscopy (SEM)

SEM visualizes morphological changes in the bacteria species. For analysis, control cells under O_3_ treatment were fixed for 1 h with 2.5% glutaraldehyde in 0.1 M cacodylate buffer. After fixation, the cells were washed three times in PBS for 5 min, post-fixed for 15 min in 1% osmium tetroxide (OsO4), and rewashed three times in PBS for 5 min. Next, the samples were dehydrated in an ascending series of ethanol (7.5, 15, 30, 50, 70, 90, and 100% ethanol) for 15 min each step, critical point dried with CO_2_, sputter-coated with a 15 nm thick layer of gold, and examined in a Jeol JSM 6390 (Tokyo, Japan) scanning electron microscope.

## 3. Results

### 3.1. Monitoring of Ozone Concentration

Monitoring the O_3_ concentration inside each container showed that the average ozone emission from the equipment (1 to 40 min) ranged from 21.1 ppm to 71.7 ppm, with the average of all measurements being 43.9 ppm (Figure 1). The mean ozone concentration in the 40 min time chosen for testing with the MDR strains was 30.8 ppm (Figure 2). The ambient temperature ranged from 22.5 °C to 24.3 °C, with an average of 23.4 °C. Regarding the relative humidity of the air, it went from 71.4% RH to 75.5% RH, with an average of 74.2% RH (Table 1, Figure 2).

### 3.2. Ozone Treatment

The culture exposure at different times (1 to 40 min) with a high level of gaseous O_3_ was able to inhibit the in vitro growth of all bacterial strains tested (Figure 1 and Figure 2) with a statistically significant reduction in colony count compared to the control group (not treated with ozone) (Table 2). Among the ATCC strains (10^5^ CFU/mL), *P. aeruginosa* (ATCC 15442) was the only one that did not significantly reduce the CFU count with only 1 min of ozone exposure, with a reduction of only 17.5% CFU. The other strains significantly reduced the number of colonies, with the most significant reduction being for *S. enterica* (ATCC 10708) (90.4%), followed by *E. coli* (ATCC 25922) and *S. aureus* (ATCC 6538) (both 98%). After 10 min of exposure to ozone, all ATCC strains showed a significant reduction in the number of counted CFU: *S. aureus* (ATCC 6538), 99.4%; *P. aeruginosa* (ATCC 15442), 93.2%; and *S. enterica* (ATCC 10708), 95.1%. *E. coli* (ATCC 25922) maintained the same percentage reduction of 98%. From 20 min to 40 min of exposure to ozone, all ATCC strains showed higher percentages of reduction in the number of CFU counts, ranging from 97.2% to 99.7%. In the MDR strains (10^5^ CFU/mL), a significant reduction in the number of counted CFU in 40 min was observed: *S. aureus* (MRSA) with a reduction of 99.99%, *P. aeruginosa* (XDR) with 99.7%, *A. baumannii* (PDR) with 99.5%, and *K. pneumoniae* (KPC+) with 95.5%. 

### 3.3. Cell Viability

Ozone treatment significantly reduced bacterial growth in *S. aureus* (MRSA), leading to an inhibition of about 99.6%, followed by *P. aeruginosa* XDR (29.2%) (Figure 3). No difference was found in bacterial viability after ozone treatment in strains of *S. aureus* (ATCC 6538), *P. aeruginosa* (ATCC 15442), *S. enterica* (ATCC 10708), *E. coli* (ATCC 25922), *A. baumannii* (PDR), and *K. pneumoniae* (KPC+) (Figure 3 and Figure 4).

### 3.4. Scanning Electron Microscopy (SEM)

Scanning electron microscopy was performed to confirm membrane damage to bacterial species. Morphological analysis showed that *S. aureus* (MRSA) and *P. aeruginosa* (XDR) present membrane alterations after O_3_ treatment. All bacterial controls showed smooth and homogeneous surfaces. The therapy produced some cell wall protrusions (Figure 5).

## 4. Discussion

Ozone generating equipment is already used as an easy and effective method of disinfection and sanitization to prevent the spread of MDR microorganisms in hospital wards. Furthermore, the portable characteristic of the equipment makes the mobile sanitation process viable for application in specific hospital areas [50,65,66]. Its high efficiency has been evaluated against many microorganisms, such as bacteria, fungi, and viruses, both on the surface and suspended in the air [45], and, for this reason, it has also been validated by several international organizations [67]. The practical applicability of ozone gas in the hospital environment can improve the microbiological condition, preventing and contributing to reducing HAI rates. For this reason, in this in vitro study, we used gaseous ozone, which has a greater disinfectant capacity due to its distribution and uniform penetration. Thus, we can inactivate microorganisms that may be present both on the surfaces and under the covers of hospital furniture [38].

Although few studies have investigated the relationship between ozone concentration and the microclimate conditions of different environments [68], some experiments have demonstrated that ozone concentration and relative humidity values played an important role in ozone efficiency and antimicrobial effect [69]. Humidity is an important parameter and must be considered because, in arid environmental conditions, the disinfection procedure may require a considerably longer exposure time. In addition, microorganisms die more quickly with increasing humidity, which favors the formation of free radicals [69]. Hudson and colleagues evaluated the effect of concentration, exposure time, and relative humidity in a study using 12 viruses. This work showed a reduction of three orders of magnitude, concerning the initial virus titer, at a concentration of 25 ppm of ozone per 15 min exposure to >90% RH [70]. Another study suggested that ozone sterilization was more effective with no air movement (no fans) at low temperature and humidity than at high temperature and humidity [71]. Finally, a recent study analyzed the influence of microclimate on the effectiveness of ozone indoors, showing that different temperature conditions, relative humidity, and distance from the ozone generator did not reduce microbial load [38]. The current study’s parameters were satisfactory, with relative humidity ranging from 71.4% RH to 77.2% RH and an average temperature of around 23.4 °C.

The total ozone dose has been considered an essential factor for biocidal activity and is calculated as the product of exposure time and concentration [72]. In 2008, Tseng and Li [73] reported that the ozone dosage required for 99% viral inactivation should be calculated as ppm × min (i.e., a product of the ozone gas concentration multiplied by the duration), obtaining a value of 114 min [ppm] at 55% relative humidity to inactivate the dsDNA virus (T7). Although it has not been tested for antiviral action on pathogenic viruses or their substitutes, Pironti et al. [38] evaluated ozone’s effectiveness in Gram-negative bacteria as an indicator of microbial contamination. By calculating the mean concentration (1.6 ppm) and the exposure time (70 min), the value of 112 min [ppm], which was very close to that suggested by others to inactivate the viruses, was obtained [73]. As stated in the literature, the critical factor for the inactivation of microorganisms in the total ozone dose is calculated as the product of the exposure time and the concentration. However, considering this calculation, our values will be higher as we use higher ozone concentrations and exposure times with large variation intervals (10 in 10 min). According to our measurements, the average ozone concentration recorded reached 43.9 ppm, reaching the minimum average value of 21.1 ppm and the maximum average value of 71.1 ppm, with the complete disappearance of the ozone after 30 min. Short exposure time to ozone was able to interfere with bacterial growth, showing that in 1 min, ozone inhibited colony growth by 90% (1 log_10_) for *S. enterica* ATCC 10708 and 98% (~2 log_10_) for *S. aureus* ATCC 6538 and *E. coli* ATCC 25922, respectively. *P. aeruginosa* ATCC 15442 was an exception, showing an inhibition rate of 17.5% with 1 min exposure to ozone. Conversely, at 10 min of exposure, its bacterial growth inhibition rate increased to approximately 95%. After exposure to ozone for 10 min, *S. aureus* ATCC 6538 showed a reduced rate in colony growth (CFU/mL) around 99% (2 log_10_). The same was verified for *S. enterica* (ATCC 10708) at 20 min and *E. coli* ATCC 25922 at 30 min. The longest exposure time used in this study was 40 min, and among the ATCC strains tested, *P. aeruginosa* ATCC 15442, despite a high value, had the lowest rate of reduction in colony growth (98.5%) in comparison to others. In the MDR strains of the ESKAPE group, ozone was able to reduce the increase by 99.99% (3 log_10_) of the colonies, followed by *P. aeruginosa* (XDR) with 99.7% (~3 log_10_), *A. baumannii* (PDR) with 99.5%, and *K. pneumoniae* (KPC+) with 95.5% (~1.5 log_10_). Our results agree with previous studies that demonstrated a reduction in colony number (CFU/mL) by around three *log_10_* bacteria known to cause hospital-acquired infections [47,49,74]. One of the studies used an ozone dose of 25 ppm for 20 min, with a short period of excess moisture (90% RH) and was able to inactivate more than 3 log_10_ in most bacteria, including *A. baumannii*, *Clostridium difficile*, and methicillin-resistant *S. aureus*, both in a laboratory test system and under simulated field conditions [49]. Another study obtained the same reduction by applying the exact ozone dosage at different exposure times and 75–95% [74]. According to Moat et al., the increase in ozone concentration can lead to disinfectant efficacy [47]. Zoutman et al. showed that it could only achieve a greater than six log_10_ reduction for MRSA at an ozone concentration of 500 ppm (exposure time 90 min) at a relative humidity of 80%, produced by a separate humidifier [75].

Reduced cell viability is one of the highly reliable biomarkers of cytotoxicity [76]. Several tests allow evaluating cell viability after a toxicity study in cultured cells. In our study, the method used to assess cell viability was the resazurin reduction assay, one of the most frequently used tests. Resazurin (7-hydroxy-3H-phenoxazin-3-one 10-oxide) is a redox dye indicator of metabolic activity in cell cultures and has numerous applications, such as toxicity, proliferation, and cell viability studies [64]. Resazurin is a non-fluorescent blue reagent that, by the action of the dehydrogenase enzyme found in metabolically active cells, is reduced to resorufin, which is highly fluorescent and has a pink color. This conversion only occurs in viable cells; as such, the amount of resorufin produced is proportional to the number of viable cells in the sample [64]. Resazurin is not toxic to cells, and the occurrence of cell death is not necessary to obtain the measurements. It is a simple and fast test that can be measured either by colorimetry or fluorimetry [62], and the amount of resorufin produced is proportional to the number of viable cells [63]. According to our results, we observed that ozone significantly reduced the in vitro growth of bacteria.

Conversely, when we investigated its metabolic capacity through resazurin, we found a significant reduction in values only for two strains, showing that ozone was able to interfere with the cell viability of *S. aureus* (MRSA), which showed inhibition of about 99.6%, followed by *P. aeruginosa* XDR (29.2%). Curiously, in a recent study using the same strains, we showed that ozone at low concentrations did not interfere with bacterial growth, but it could significantly inhibit cell viability [51]. Interestingly, all species’ reference strains (ATCC) were less susceptible to ozone treatment. Similarly, a study demonstrated that the antibiotic resistance of the isolates was not correlated to a higher ozone tolerance [77]. The increased susceptibility of PaXDR and MRSA to ozone may be due to a metabolic cost associated with antibiotic resistance that decreases fitness and reduces the ecological versatility of resistant strains [78].

Although not as pronounced, the effectiveness of ozone as a disinfectant varies significantly between different types of bacteria, even at the strain level [79,80], and depends on several factors such as the growth stage, the cell envelope, the efficiency of repair mechanisms, and the type of viability indicator used [81,82,83]. In addition, some factors can reduce the ozone stability or protect microorganisms from its effects, thus decreasing the disinfection efficiency, such as concentration and type of dissolved organic material or the presence of flakes or particles [84,85,86]. Yet, ozone decomposition results in superoxide, hydroperoxyl, and hydroxyl radicals [87,88]. Microorganisms, through detoxification enzymes, can develop mechanisms such as the production of superoxide dismutases, reductases, peroxides, and catalases to neutralize the lethal effects of reactive oxygen species [58,89,90]. In *E. coli*, two of these mechanisms’ (*SoxR* and *OxyR*) responsive redox transcription regulators have already been well described [91]. Both regulators are induced in the presence of radicals [92] and activate several genes such as *soxS* and *sod*, which, in turn, protect against these radicals through DNA repair or removal of the radicals [91]. *DnaK* and *RpoS* are two general stress gene regulators that, although not dedicated mechanisms of protection against oxidative radicals, have previously been shown to confer protection against them [93,94,95]. *S. aureus* uses the expression of several of these detoxification proteins, including catalase (*katA*), superoxide dismutase (*sodA*, *sodM*), thioredoxin reductase (*trxB*), thioredoxin (*trxA*), alkyl hydroperoxide reductase (*ahpC*, *ahpF*) enzymes, and glutathione peroxidase (*gpxA*) [96]. Similar radicals are produced during ozone treatments; therefore, these genes are expected to play an important role in protecting cells against this technology in different bacteria that could also justify interfering with cell viability.

The disinfectant potential of ozone is attributed to its ability to promote cell wall disturbance and extravasation of ions and intracellular molecules, triggering cell death [96]. The primary cellular targets for ozone are nucleic acids, where damage can range from base lesions to single- and double-strand breaks [80]. Lesions can lead to more or less compromising point mutations, whereas massive DNA breakage is lethal if not repaired [96,97,98,99]. Many studies prove that the cell envelope is also affected during ozonation, even before severe DNA damage [100,101,102]. Ozone can influence the global polarity of the bacterial surface [58], involving mechanisms of lipid peroxidation [59,103] and the degradation of transmembrane proteins that control the flow of ions. As a result, the cells will rupture with a subsequent leakage of ions between the media, resulting in the death of the microorganism [60]. In addition, the high oxidative potential of ozone contributes to changes in the zeta potential. A physical property is applied to assess the degree of peripheral electronegativity on the cell surface when suspended in a fluid [104]. In a study by Feng et al. (2018), as the ozone dose increased, the zeta potential tended to decrease, becoming hostile and causing greater bacterial instability in the medium [58,105]. Ozone is a gas that can oxidize glycoproteins, glycolipids, and cell wall amino acids, destroying sulfhydryl groups in enzymes and causing the breakdown of cell enzymatic activity [106,107].

Our study expands on and corroborates what is already known about the gas since the analysis of the inhibition of microbial growth and/or reduction of the CFU count in plates exposed to ozone, containing both reference strains and clinical and environmental strains highly resistant to antimicrobials, compared to the control group, proved its effectiveness as a chemical compound in microbial control processes. The practical applicability of gaseous ozone in hospital environments can improve the microbiological condition, preventing and contributing to HAI rates. It is exciting and unprecedented evidence of the potential for ozone disinfection because, in natural indoor environments, it is possible to disinfect surfaces not typically disinfected with hand-applied liquid disinfectants. In this sense, it can eliminate MDR organisms with a significant advantage compared to mechanical disinfection methods with liquid disinfectants of environmental surfaces in health care establishments, including the hospital environment, where it is common to use other chemical compounds in liquid form.

## 5. Conclusions

HAIs represent the most common adverse event in ICUs and are usually caused by MDR bacteria. As a result, preventing the transmission of MDR bacteria has become increasingly important to limit the spread of these infections, and a correct sanitization protocol is particularly crucial. Given the prolonged hospital stays and increased treatment costs seen in patients who develop HAIs, ozone-based decontamination approaches that have low toxicity compared to other sanitization products may represent the future of hospital cleaning methods as a highly cost-effective and promising intervention capable of being used as an additional procedure for terminal cleaning, in addition to the “classic” terminal cleaning (by current biocides). Our results evidenced the antimicrobial potential of gaseous ozone in bacteria that are currently a significant problem worldwide. In the future, this resource may be a part of the protocol for the disinfection of hospital environments and surfaces, ensuring the control of microbial development.

## Figures and Tables

**Figure 1 molecules-27-03998-f001:**
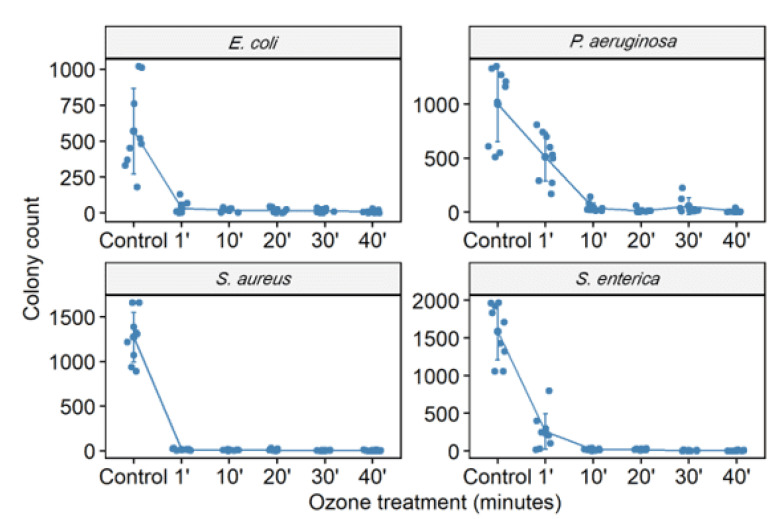
The number of colony-forming units (CFU) in different bacterial strains (*S. aureus* (ATCC 6538), *P. aeruginosa* (ATCC 15442), *S. enterica* (ATCC 10708), and *E. coli* (ATCC 25922)) was counted. CFU counting was performed in the control group (no treatment) and bacterial suspensions (10^5^ CFU/mL) after exposure to ozone for 1, 10, 20, 30, and 40 min.

**Figure 2 molecules-27-03998-f002:**
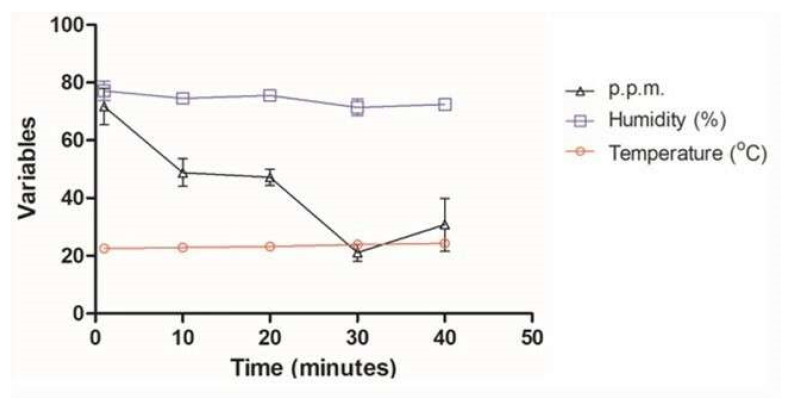
Average temperature, relative humidity, and ozone concentration at times of 1, 10, 20, 30, and 40 min with ATCC (*S. aureus* (ATCC 6538), *P. aeruginosa* (ATCC 15442), *S. enterica* (ATCC) strains 10708), and *E. coli* (ATCC 25922)) and multidrug-resistant *S. aureus* (MRSA), *P. aeruginosa* (XDR), *A. baumannii* (PDR), and *K. pneumoniae* (KPC+).

**Figure 3 molecules-27-03998-f003:**
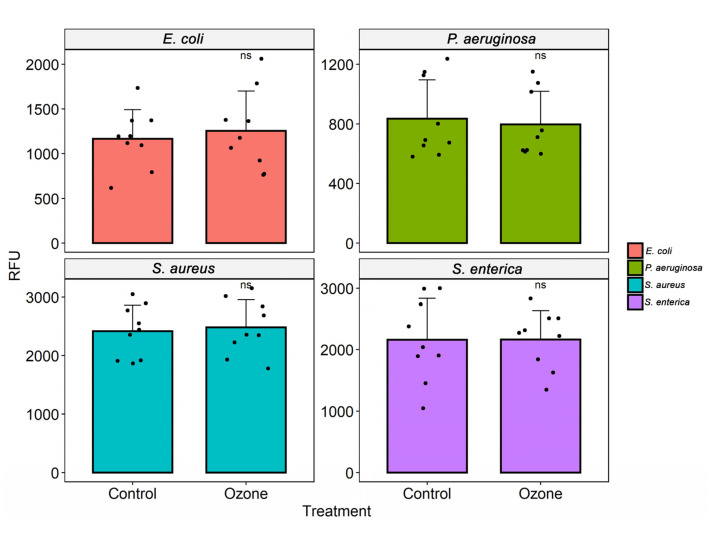
Analysis of cell viability after ozone treatment in different bacterial strains (*S. aureus* (ATCC 6538), *S. enterica* (ATCC 10708), *E. coli* (ATCC 25922), and *P. aeruginosa* (ATCC 15442)). The measurement of fluorescence intensity (relative fluorescence units, RFU) after the conversion of resazurin to resofurin by viable bacteria was performed in the control group (no treatment) and bacterial suspensions (10^5^ CFU/mL) after exposure to ozone for 40 min. Results represent values from 3 randomly chosen colonies in the control group (no treatment) and after treatment with ozone. The black dots represent the values of fluorescence emission after addition of resazurin.

**Figure 4 molecules-27-03998-f004:**
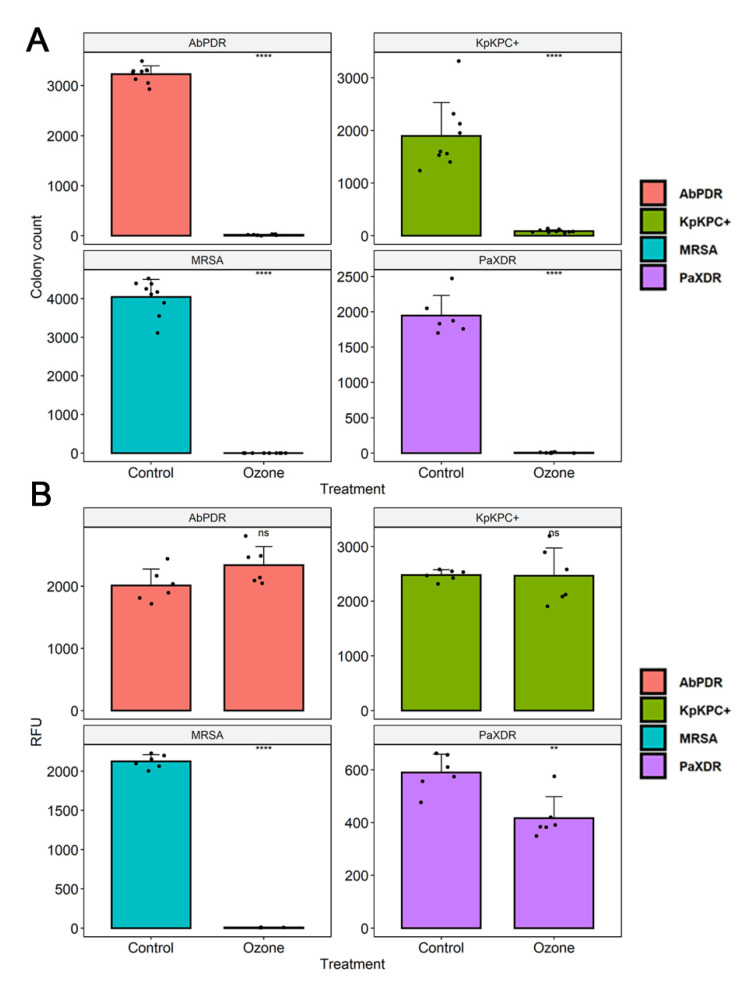
(**A**) The number of colony-forming units (CFU) in different bacterial strains (*S. aureus* (MRSA), *P. aeruginosa* (XDR), *A. baumannii* (PDR), and *K. pneumoniae* (KPC+)) were counted. The number of CFU in the control group (no treatment) and bacterial suspensions (10^5^ CFU/mL) after exposure to ozone for 40 min was quantified. The black dots represent the number of CFU count of the different strains. (**B**) Analysis of cell viability after ozone treatment in different bacterial strains (*S. aureus* (MRSA), *P. aeruginosa* (XDR), *A. baumannii* (PDR), and *K. pneumoniae* (KPC+)). The measurement of fluorescence intensity (Relative fluorescence units, RFU) after the conversion of resazurin to resofurin by viable bacteria was performed in the control group (no treatment) and bacterial suspensions (10^5^ CFU/mL) after exposure to ozone for 40 min. Results represent three randomly chosen colonies in the control group (no treatment) and after ozone treatment. The black dots show cell viability values through fluorescence emission after the addition of resazurin. ** Statistically significant (*p* < 0.01); **** statistically significant (*p* < 0.001).

**Figure 5 molecules-27-03998-f005:**
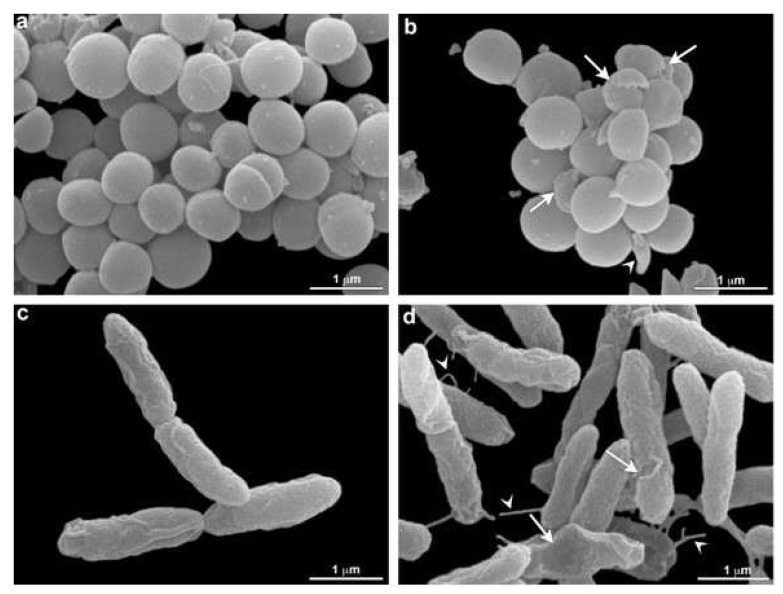
Morphological analysis of O_3_ treatment by electron microscopy. *S. aureus* (MRSA) (**a**,**b**) and *P. aeruginosa* (XDR) (**c**,**d**) are seen without (**a**,**c**) and under O_3_ treatment (**b**,**d**). An alteration in *S. aureus* (MRSA) shape is seen (arrowhead) in b. Damage in bacteria is observed after treatment (arrows) (**b**). Note that control cells are rounded and present in a homogeneous surface (**a**). Damaged cells are observed after treatment in *P. aeruginosa* (XDR) (arrows) (**d**). Some cell wall protrusions are observed in treated cells (arrowhead) (**d**). These aspects were not verified in control cells (**c**).

**Table 1 molecules-27-03998-t001:** Monitoring ozone concentration, temperature, and humidity of different bacterial strains after exposure to ozone.

Ozone Exposure Time (Minutes)	Evaluated Parameters		
	Temperature (°C)	Relative Humidity of Air (% RH)	Ozone Concentration (ppm)
1	22.5	77.2	71.7
10	22.8	74.6	48.8
20	23.7	75.5	47.2
30	23.9	71.4	21.1
40	24.3	72.4	30.8
Mean	23.4	74.2	43.9

**Table 2 molecules-27-03998-t002:** Count and percentage reduction in the number of CFU in ATCC (*S. aureus* (ATCC 6538), *P. aeruginosa* (ATCC 15442), *S. enterica* (ATCC 10708), and *E. coli* (ATCC 25922)) strains and multidrug-resistant *S. aureus* (MRSA), *P. aeruginosa* (XDR), *A. baumannii* (PDR), and *K. pneumoniae* (KPC+). ATCC strains (10^5^ CFU/mL) after exposure to ozone (1, 10, 20, 30, and 40 min) and in bacterial suspensions of multi-resistant strains (10^5^ CFU/mL) after exposure to ozone (40 min).

Bacterial Strains	Ozone Exposure Times					
	C	1′	10′	20′	30′	40′
		Count Number CFU/% of Reduction				
*S. aureus* (ATCC 6538)	6287	123.1/98	36/99.4	31.9/99.5	27.2/99.6	20.4/99.7
*P. aeruginosa* (ATCC 15442)	3767	3109/17.5	256/93.2	105.2/97.2	65.33/98.3	57.8/98.5
*S. enterica* (ATCC 10708)	7391	711.3/90.4	360.8/95.1	69.7/99.1	63.11/99.2	53.9/99.3
*E. coli* (ATCC 25922)	3090	62/98	66.1/98	57.2/98.1	31.2/99	30.3/99
*S. aureus* (MRSA)	4041	-	-	-	-	0.1/99.99
*P. aeruginosa* (XDR)	1946	-	-	-	-	6.33/99.7
*A. baumannii* (PDR)	3228	-	-	-	-	16.6/99.5
*K. pneumoniae* (KPC+)	1894	-	-	-	-	86/95.5

C: control not exposed to ozone; CFU: colony-forming units.

## Data Availability

The data presented in this study are available upon request from the corresponding author.

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
