# Peer review of "Potent Activity of a High Concentration of Chemical Ozone against Antibiotic-Resistant Bacteria"

_molecules, 2022, doi:10.3390/molecules27133998_

Round 1
Reviewer 1 Report
The paper by Rangel et al., put the term therapy in the title and according to this reviewer this is incorrect, as the environmental and chemical use of ozone is not a therapy. I recommend the authors to revise the title, accordingly, for example as follows: “Potent Activity of High Concentration of Chemical Ozone Against Antibiotic-Resistant Bacteria”. The term therapy must be removed.
Introduction:
Lines 42-44: developing countries….developed countries…Please try to find synonymous in order to avoid repetitions.
Regarding HAI I would add a sentence or context regarding the role of HAI in increasing hospitalization-associated deaths during the COVID-19 pandemic (add these refs Pandolfi S, Valdenassi L, Bjørklund G, Chirumbolo S, Lysiuk R, Lenchyk L, DoÅŸa MD, Fazio S. COVID-19 Medical and Pharmacological Management in the European Countries Compared to Italy: An Overview. Int J Environ Res Public Health. 2022 Apr 2;19(7):4262 and Chirumbolo S, Simonetti V, Franzini M, Valdenassi L, Bertossi D, Pandolfi S. Estimating coronavirus disease 2019 (COVID-19)-caused deaths in hospitals and healthcare units: Do hospital-acquired infections play a role? Comments with a proposal. Infect Control Hosp Epidemiol. 2021 Mar 19:1-2).
Author Response
1). The paper by Rangel et al. put the term therapy in the title, and according to this reviewer, this is incorrect, as the environmental and chemical use of ozone is not a therapy. I recommend the authors revise the title accordingly, for example: "Potent Activity of High Concentration of Chemical Ozone Against Antibiotic-Resistant Bacteria ."The term therapy must be removed.
Thanks. The title was revised.
Introduction:
2). Lines 42-44: developing countries….developed countries…Please try to find synonymous to avoid repetitions.
Thanks. It was done.
3). Regarding HAI, I would add a sentence or context regarding the role of HAI in increasing hospitalization-associated deaths during the COVID-19 pandemic (add these refs Pandolfi S, Valdenassi L, Bjørklund G, Chirumbolo S, Lysiuk R, Lenchyk L, DoÅŸa MD, Fazio S. COVID-19 Medical and Pharmacological Management in the European Countries Compared to Italy: An Overview. Int J Environ Res Public Health. 2022 Apr 2;19(7):4262 and Chirumbolo S, Simonetti V, Franzini M, Valdenassi L, Bertossi D, Pandolfi S. Estimating coronavirus disease 2019 (COVID-19)-caused deaths in hospitals and healthcare units: Do hospital-acquired infections play a role? Comments with a proposal. Infect Control Hosp Epidemiol. 2021 Mar 19:1-2).
Thanks for the suggestion. It was inserted in lines 81-88.

Reviewer 2 Report
The authors presented the paper "Potent Activity of High Concentration Ozone Therapy Against Antibiotic-Resistant Bacteria"
1) The more fresh (2-3 years) references should be included in the introduction section. Moreover, 2022 year papers related to the topic of the work should be added to the reference list.
2) It would be nice for the authors focus on their problems in introduction section and to shorten it a bit. For example, everybody knows the properties of O3, but it will be better not to discuss its plant application but to present relevant papers about antibacterial activity. Moreover, the information about the a large number of drug-resistance bacteria is publicly available. The 78 references for the experimental paper is too high.
3) I understand that O3 is a highly reactive compound and good oxidizing agents. In this way, the novelty of the paper material should be clearly mentioned in the Abstract and Conclusion section.
4) Extensive discussion of Figures 3 and 4 should be inserted. Moreover, I have seen the link on Figure 3 in the line 307. That is why Figure 3 should be presented after the link.
5) In discussion section I see a lot of literature information, but not the information from the authors' results. I think that literature data have to be shortened.
Minor comments
1) Table 1 and Table in Figure 2. It looks like that it is a picture. And the picture is low quality. The same problem with low quality with Figures 3 and 4.
2) , should be changed to . in Table 1 and elsewhere.
Author Response
The authors presented the paper "Potent Activity of High Concentration Ozone Therapy Against Antibiotic-Resistant Bacteria."
1) The more fresh (2-3 years) references should be included in the introduction section. Moreover, 2022 year papers related to the topic of the work should be added to the reference list.
Thanks. More new references were included. For example, line 132, lines 129-135, 141-147,
and 154-157.
2) It would be nice for the authors to focus on their problems in the introduction section and shorten them. For example, everybody knows the properties of O3, but it will be better not to discuss its plant application but to present relevant papers about the antibacterial activity. Moreover, the information about many drug-resistance bacteria is publicly available. Finally, the 78 references for the experimental article are too high.
Thanks. The antibacterial activity was presented in lines 148-154. The references were shortened.
3) I understand that O3 is a highly reactive compound and a good oxidizing agent. In this way, the novelty of the paper material should be clearly mentioned in the Abstract and Conclusion section.
Thanks. It was mentioned. Lines 35-37and Lines 492-498.
4) Extensive discussion of Figures 3 and 4 should be inserted. Moreover, I have seen the link in Figure 3 in line 307. That is why Figure 3 should be presented after the link.
Thanks. Figure 3 was moved after the link.
5) In the discussion section, I see a lot of literature information, but not the information from the authors' results. Therefore, I think that literature data have to be shortened.
Thanks. It was shortened.
Minor comments
6) Table 1 and Table in Figure 2. It looks like it is a picture. And the picture is low quality. The same problem is with low quality in Figures 3 and 4.
7) should be changed to. In Table 1 and elsewhere.
All images were sent to the editor with a resolution of 600 dpi. However, when converted to pdf, they lose quality. The original file has adequate resolution and quality.

Round 2
Reviewer 2 Report
Thank you for the revised paper.